# Evolution of the *vls* Antigenic Variability Locus of the Lyme Disease Pathogen and Development of Recombinant Monoclonal Antibodies Targeting Conserved VlsE Epitopes

Li Li,[a] Lia Di,[b] Saymon Akther,[a] Brian M. Zeglis,[a,c,d,e] Weigang Qiu[a,b,f,g]

[a]Graduate Center, City University of New York, New York, New York, USA
[b]Department of Biological Sciences, Hunter College, City University of New York, New York, New York, USA
[c]Department of Chemistry, Hunter College, City University of New York, New York, New York, USA
[d]Department of Radiology, Weill Cornell Medical College, New York, New York, USA
[e]Department of Radiology, Memorial Sloan Kettering Cancer Center, New York, New York, USA
[f]Department of Physiology and Biophysics, Weill Cornell Medical College, New York, New York, USA
[g]Institute for Computational Biomedicine, Weill Cornell Medical College, New York, New York, USA

**ABSTRACT** VlsE (variable major protein-like sequence, expressed) is an outer surface protein of the Lyme disease pathogen (*Borreliella* species) responsible for its within-host antigenic variation and a key diagnostic biomarker of Lyme disease. However, the high sequence variability of VlsE poses a challenge to the development of consistent VlsE-based diagnostics and therapeutics. In addition, the standard diagnostic protocols detect immunoglobins elicited by the Lyme pathogen, not the presence of the pathogen or its derived antigens. Here, we described the development of recombinant monoclonal antibodies (rMAbs) that bound specifically to conserved epitopes on VlsE. We first quantified amino-acid sequence variability encoded by the *vls* genes from 13 *B. burgdorferi* genomes by evolutionary analyses. We showed broad inconsistencies of the sequence phylogeny with the genome phylogeny, indicating rapid gene duplications, losses, and recombination at the *vls* locus. To identify conserved epitopes, we synthesized peptides representing five long conserved invariant regions (IRs) on VlsE. We tested the antigenicity of these five IR peptides using sera from three mammalian host species including human patients, the natural reservoir white-footed mouse (*Peromyscus leucopus*), and VlsE-immunized New Zealand rabbits (*Oryctolagus cuniculus*). The IR4 and IR6 peptides emerged as the most antigenic and reacted strongly with both the human and rabbit sera, while all IR peptides reacted poorly with sera from natural hosts. Four rMAbs binding specifically to the IR4 and IR6 peptides were identified, cloned, and purified. Given their specific recognition of the conserved epitopes on VlsE, these IR-specific rMAbs are potential novel diagnostic and research agents for direct detection of Lyme disease pathogens regardless of strain heterogeneity.

**IMPORTANCE** Current diagnostic protocols of Lyme disease indirectly detect the presence of antibodies produced by the patient upon infection by the bacterial pathogen, not the pathogen itself. These diagnostic tests tend to underestimate early-stage bacterial infections before the patients develop robust immune responses. Further, the indirect tests do not distinguish between active or past infections by the Lyme disease bacteria in a patient sample. Here, we described novel monoclonal antibodies that have the potential to become the basis of direct and definitive diagnostic detection of the Lyme disease pathogen, regardless of its genetic heterogeneity.

**KEYWORDS** *Borrelia burgdorferi*, C6 peptide, Lyme disease, VlsE, monoclonal antibodies

Lyme disease is a multistage, tick-transmitted infection caused by spirochetes of the bacterial species complex *Borrelia burgdorferi sensu lato* (*Bbsl*), known more concisely

Address correspondence to Weigang Qiu, wqiu@hunter.cuny.edu.

The authors declare no conflict of interest.

(albeit controversially) as a new genus *Borreliella* (1, 2). Lyme disease is the most common tick-borne disease in regions of North America, Europe, and Asia (3, 4). In the United States, approximately 476,000 cases are diagnosed annually (5). Most Lyme disease cases in the US are caused by the single species *B. burgdorferi* and transmitted by the hard-bodied *Ixodes scapularis* or *I. pacificus* ticks, although the same tick vectors carry other *Borreliella* species as well as *Borrelia* species closely related to relapsing fever spirochetes (6–8). *B. burgdorferi* causes multisystemic manifestations in humans including erythema migrans (EM) at early stages, arthritis, carditis, neuroborreliosis in late stages, and chronic symptoms associated with persistent infections (4, 9, 10).

Antigenic variation via continuously altering the sequences of surface antigens during infection is a common strategy that microbial pathogens employ to escape the adaptive immune responses of vertebrate hosts (11, 12). In the two sister spirochetal groups – *Borrelia* causing relapsing fever and *Borreliella* causing Lyme disease, two homologous but distinct molecular systems have evolved facilitating continuous antigenic variation through recombination between an expressed locus and silent archival loci during persistent infection within the vertebrate hosts (13). In *B. burgdorferi*, the molecular system able to generate antigenic variation consists of one expression site (*vlsE*, variable major protein-like sequence, expressed) and a set of tandemly arranged silent cassettes that share more than 90% similarities to the central cassette region of *vlsE* (14–17) (Fig. 1). During mammalian infection, *vlsE* continuously expresses and undergoes random segmental recombination with the silent cassettes, generating a considerable number of new VlsE antigen variants to prolong spirochete infection in hosts (13, 16).

The *vlsE* gene encodes a 36 kDa lipoprotein that is anchored to the outer membrane on the cell surface. The primary structure of VlsE comprises the N- and C-terminal domains, as well as the central cassette, which consists of six highly variable regions (VR1-VR6), interspersed with six conserved invariant regions (IR1-IR6) (Fig. 1). The N- and C-terminal regions do not undergo antigenic variation and are thought to be important in maintaining the functional structure of the molecule (15). The cassette sequences undergo antigenic variation during infection (18). The crystal structure of recombinant VlsE protein revealed that the six VRs constitute loop structures and form a "dome" on the membrane distal surface exposed to the host environment, which may shield the IRs from antibody binding (19).

VlsE elicits strong humoral responses that can be detected throughout Lyme disease, making it a powerful antigen in serologic assays of Lyme disease diagnosis (18–21). Contrary to the established paradigm of weak immunogenicity of the conserved regions of bacterial surface proteins, the conserved IR6 elicits immunodominant antibody responses during human infection despite the region being largely inaccessible on the intact VlsE molecule (18, 22, 23). The surprising finding of immunodominance of IR6 in human patients is hypothesized to be a result of antigen processing of the VlsE proteins in nonreservoir host species (24).

A 26-amino acid peptide that reproduces the IR6 sequence, known as the C6 peptide, is used in commercial diagnostic tests for Lyme disease (18, 25). The standardized two-tiered testing (STTT) for Lyme disease diagnosis includes a screening enzyme immunoassay (EIA) with the whole-cell sonicate and a subsequent confirmatory Western blot assay for the presence of both IgM and IgG antibodies against 10 *Borreliella* antigens (26, 27). Recently, a modified two-tiered testing (MTTT) protocol using two sequential EIAs with C6 peptide or the whole VlsE protein has been developed. MTTT improved sensitivity and specificity relative to STTT, especially in Lyme patients with early-stage manifestations (28). Nevertheless, the overall sensitivity for early-stage diagnosis remains low, ranging from 36% to 54%, even with MTTT (29). In addition, both diagnostic assays are indirect tests and do not distinguish between active infection and past exposure. In summary, there is a need to simplify the testing protocol for Lyme disease, improve testing sensitivity in the early infection stage, and detect the presence of the Lyme pathogen or its derivative antigens directly.

During the transmission cycle of *B. burgdorferi*, the *vls* locus is expressed during the late-stage persistent infection within the mammalian host, in contrast to genes like

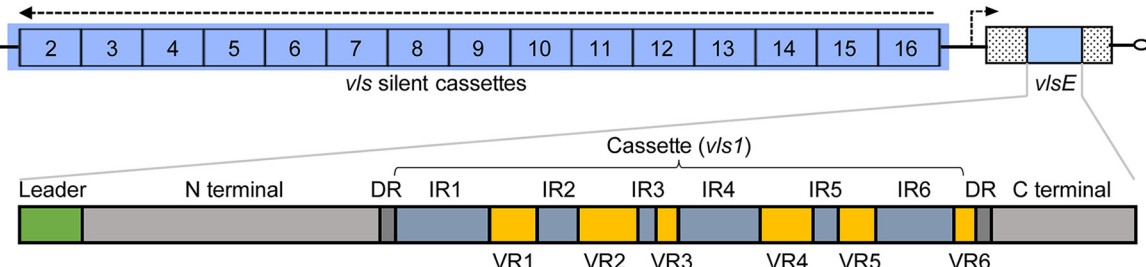

**FIG 1** Genomic and gene structures of the *vls* locus in *B. burgdorferi* strain B31. (Top) The *vls* locus is located close to the telomere of the linear plasmid lp28-1 (GenBank accession AE000794) in the B31 genome, consisting of cassettes of silent (unexpressed) open reading frames (ORFs) (*vls2* through *vls16*) and an expressed ORF (*vlsE*) containing the *vls1* cassette introduced by recombination (14). Dashed arrows indicate the direction of coding strands. (Bottom) The VlsE protein consisted of a leader peptide, an N terminus domain, a cassette flanked by two direct repeats (DRs), and a C terminus domain. The central cassette consisted of interspersed variables (VR1-6) and invariant regions (IR1-6).

*ospA* (encoding outer surface protein A) expressed within the ticks, and genes like *ospC* expressed exclusively during a short window of time when the spirochetes begin to migrate from the tick to the mammalian host (30–32). As a multicopy gene family and driven by diversifying natural selection, the silent *vls* cassettes exhibit high sequence variability not only between *B. burgdorferi* strains but also within the same genome (33–35). In the present study, we developed a bioinformatics workflow to facilitate the automated identification of *vls* sequences from the sequenced *Borreliella* genomes. We quantified evolutionary rates at individual amino acid sites of the *vls* coding sequences identified from 13 *B. burgdorferi* genomes. Extending the previous analysis of mechanisms of evolution at the *vls* locus (8, 34), we explored the evolution mechanisms by comparing the *vls* gene phylogeny with the genome-derived strain phylogeny. Our experimental investigations of the immunogenicity of the VlsE protein confirmed the immunodominance of the IR6 peptide and discovered a similar immunodominance of the IR4 peptide in human patients and immunized rabbits but not the reservoir hosts. Furthermore, we identified, cloned, and purified four recombinant IR-specific monoclonal antibodies (rMAbs) that are promising theragnostic agents for the direct assay of *B. burgdorferi* infection in clinical samples and model organisms of Lyme disease.

## RESULTS

**Phylogenetic inconsistencies indicated duplications and losses, sequence divergence, and recombination at the *vls* locus.** We identified 194 *vls* cassette sequences from 13 *B. burgdorferi* strains and inferred a maximum likelihood tree of the cassette (Fig. 2). These *B. burgdorferi* strains have been classified into four phylogenetic groups (A to D) based on chromosomal single-nucleotide polymorphisms (SNPs) (36). The *vls* gene phylogeny consists of eight major clades and is consistent with a previously published *vls* cassette phylogeny (34). Here, we analyzed the *vls* gene phylogeny in the broader context of strain phylogeny. Phylogenetic inconsistencies between gene and strain trees may result from – and thus indicate the occurrence of – horizontal gene transfers between strains, ancestral gene duplications followed by the loss of duplicated copies, and incomplete lineage sorting when strains rapidly diverge from one other (37, 38).

The *vls* sequences from the two SNP group D strains (JD1 and 156a) formed a monophyletic group consistent with the strain phylogeny. However, within this major clade, the *vls* sequences did not separate into two strain-specific clades. This phylogenetic inconsistency could be caused by a mixture of paralogous and orthologous gene copies as a result of random gene duplications and losses. Another possible cause of the mixed paralogs from two closely related strains is incomplete lineage sorting, by which descendant strains stochastically inherited gene copies from a common ancestor (39). Horizontal gene exchanges, on the other hand, are unlikely the reason for this inconsistency because recombination would have introduced *vls* sequences from outside the SNP group D.

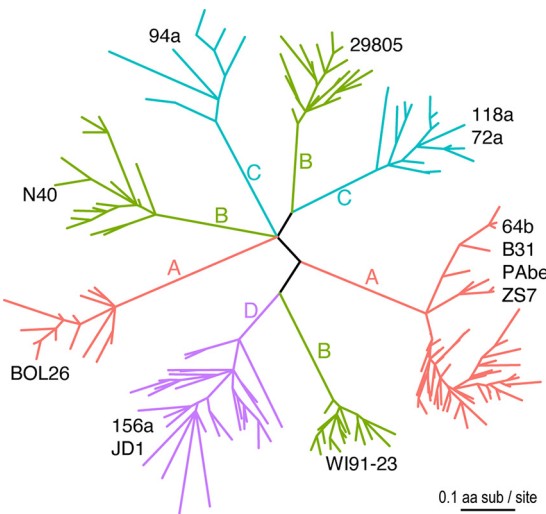

**FIG 2** Sequence diversity of *vls* cassettes of *B. burgdorferi* strains. Eight major clades of *vls* alleles were identified based on the codon alignment of 194 cassette sequences from 13 North American and European *B. burgdorferi* genomes (strain names shown by the clades) (35). The maximum likelihood tree was inferred using IQ-TREE (version 1.6.1) (71). All branches were supported by ≥80% bootstrap values. The tree was rendered using the R package *ggtree* (Version 2.2.4) (72). Branches were colored according to the four phylogenetic groups (A through D) identified based on genome-wide single-nucleotide polymorphisms (SNPs). SNP groups A, B, and C split into multiple clades, indicating rapid *vls* sequence divergence between closely related strains (34).

In contrast, the *vls* sequences from the strains belonging to the SNP groups A, B, and C all formed paraphyletic groups, each of which contained multiple clades highly divergent from one another than one would expect from the strain phylogeny (Fig. 2). In the SNP group A, the *vls* sequences from the European strain BOL26 formed a clade highly divergent from the *vls* sequences from the North American strains B31, PAbe and 64B and the European strain ZS7. In the SNP group B, the *vls* sequences from three strains (WI91-23, N40, and 29805) formed three strain-specific clades. The *vls* sequences from strains belonging to the SNP group C were split into two clades, one consisting of the sequences from strain 94a and the other consisting of sequences from strains 72a and 118a.

As in group D, the *vls* sequences within the SNP groups A, B, and C did not sort into strain-specific clades, indicating similar evolutionary processes including frequent gene duplications, rapid gene losses, and fast sequence divergence within each group. Indeed, it has been shown that the rapid sequence evolution of the *vls* cassettes was driven by adaptive differentiation evidenced by the accelerated nonsynonymous nucleotide substitutions (i.e., a high *dN/dS* ratio) (34).

**Evolutionary rates and molecular structure of *vls* cassettes.** Rates of amino-acid substitutions are not uniform along the translated *vls* sequence, which consists of mostly fast-evolving variant regions (VRs) interspersed with six short conserved invariant regions (IR1-6) (14). Here, we quantified *vls* variability at individual amino-acid sites among the 13 *B. burgdorferi* strains using the 194 *vls* sequences including both the expressed and unexpressed cassettes. Conserved regions were detected by computing the relative evolutionary rate of each amino-acid site in the multiple sequence alignment, with the average variability score scaled to zero (Fig. 3). Most residues in the IRs showed negative variability scores, indicating below-average evolutionary rates. The mean variability score for each IR was shown in Table 1.

We further mapped the IRs to a published three-dimensional structure of the VlsE protein (from the strain B31) (19) (Fig. 4). All the IRs formed alpha helixes, as the ribbon model showed (Fig. 4A). The space-filled model showed that IR1, IR2, and IR4 were partially surface exposed while IR3, IR5, and IR6 exhibited limited surface exposure (Fig. 4B). The VlsE molecules likely form dimers on the spirochete cell surface (16, 19,

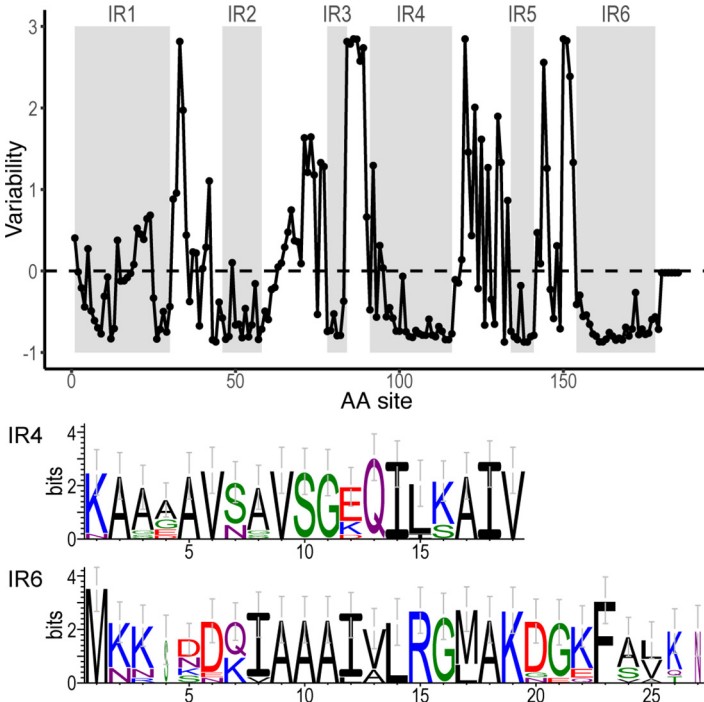

**FIG 3** Site-specific evolutionary rates of *vls* cassettes. (Top) Evolutionary rate, denoted as variability score (in the unit of standard deviation, *y*-axis), at each amino acid site was estimated by Rate4Site (Version 3.0.0) (73) based on an alignment of translated sequences of 194 *vls* cassettes and the maximum-likelihood tree (Fig. 2). The dashed line at 0 indicates the average evolutionary rate. The six IRs, showing generally lower-than-average rates, were shaded in gray. VlsE of the B31-5A3 clone (GenBank accession U76405) was used as the reference for computation and annotation. (Bottom) SeqLogo images of IR4 and IR6 sequences, constructed based on 12 representative *vls* alleles (translated) (35). Amino acid residues were colored according to physiochemistry. Letter heights correspond to information content in bits, a measure of site conservation (74).

40, 41), which would further shield the invariant regions located on the monomer-monomer interface (Fig. 4C and D). Nevertheless, IR4 is partially exposed at the membrane distal surface even in a dimerized form (Fig. 4D).

**Antigenicity of IRs against host sera.** We measured the antigenicity of IRs with sera from Lyme disease patients, white-footed mice, and recombinant VlsE-immunized rabbits using an enzyme-linked immunosorbent assay (ELISA). For the 46 human sera, reactivities of the IR4 and IR6 peptides with the human sera were significantly higher than that of BSA ($P$ = 2.87e−13 and 3.06e−13 by analysis of variance [ANOVA], respectively), while reactivities of IR1, IR2, and IR5 were less significant ($P$ = 0.034, 0.034, and 0.0019 by ANOVA, respectively) (Fig. 5A and B, left). Reactivity of VlsE recombinant pro-

**TABLE 1** Peptides used to screen for IR-specific monoclonal antibodies

| Peptide[a] | Sequence[b] | Length | Variability (*z*-score)[d] |
|---|---|---|---|
| IR1 | EVSELLDKLVKAVKTAEGASSG | 22 | −0.1746 |
| IR2 | (ASVK)GIAKGIKEIVEAA(GGSE) | 21 | −0.6066 |
| IR3[c] | AGKLFVK | 7 | −0.6585 |
| IR4 | KAAGAVSAVSGEQILSAIV(TAA) | 22 | −0.5393 |
| IR5 | (AEEAD)NPIAAAIG(TTNEDA) | 19 | −0.7378 |
| IR6 | MKKDDQIAAAIALRGMAKDGKFAVK | 25 | −0.6932 |

[a]IRs: Invariant regions. Six IRs were identified from an alignment of VlsE proteins from three strains representing two species (18).

[b]IR sequences were based on the VlsE protein of the B31 strain (GenBank accession U76405). Flanking residues (in parentheses) were padded to the IR2 and IR4 to facilitate ELISA. For IR5, the padded residues were the most common residues flanking this conserved region.

[c]Antigenicity of IR3, the shortest IR, was not tested in the present study.

[d]Standard deviation from the mean amino-acid substitution rate of zero, with a lower score indicating a higher level of sequence conservation (see Materials and Methods).

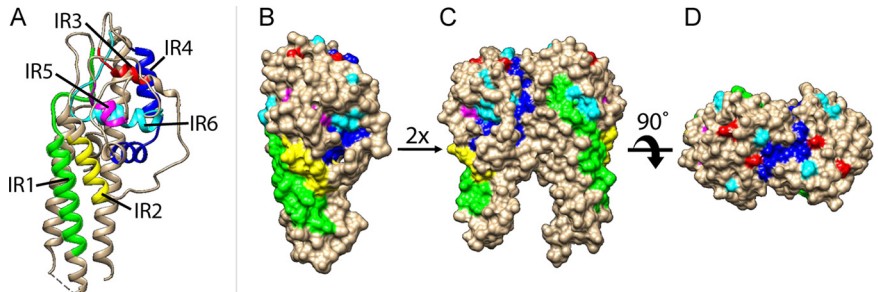

**FIG 4** IR-highlighted three-dimensional structures of VlsE. Structure diagrams of VlsE protein from B31 were prepared in Chimera (Version 1.15) (77) based on the PDB file (accession 1L8W) (19). IRs were highlighted in different colors (IR1 in green, IR2 in yellow, IR3 in red, IR4 in dark blue, IR5 in magenta, and IR6 in cyan). (A) Ribbon diagram showing that IRs tend to form alpha helices. (B) Surface-filled diagram showing membrane surface exposure of IRs in monomeric form. (C) and (D) Dimerized structure models. The structures were oriented to show the membrane-proximal part at the bottom (A, B, and C).

tein was the strongest ($P < 2.2e-16$ by ANOVA). In addition, reactivities of IR4 and IR6 with the human sera were weakly although significantly correlated with those of the VlsE ($P = 7.6e-4$ and $R^2 = 0.212$ for IR4, $P = 3.6e-3$ and $R^2 = 0.158$ for IR6, both by linear regression). Reactivities of both early and late-stage samples were significantly higher than the non-Lyme control samples ($P = 8.49e-5$ and $P = 2.63e-4$, respectively, by $t$ test) but there was no significant difference in reactivity between the early and late-stage patient samples ($P = 0.8654$ by $t$ test).

For sera from 10 naturally infected white-footed mice, the natural reservoir host of *B. burgdorferi*, the IR peptides showed weakly significant differences in antigenicity among the antigens ($P = 0.0159$ by ANOVA), with only VlsE showing a significant difference from the BSA control ($P = 2.9e-3$ by ANOVA) (Fig. 5B, middle). These results are consistent with the findings of an earlier study that showed low antigenicity of the IR6 peptide in natural hosts relative to its antigenicity in humans (18).

The gross anti-VlsE polyclonal antibodies were extracted from the serum of four immunized rabbits. Reactivities of the IRs against the rabbit polyclonal antibodies showed a similar pattern as those against the naturally infected human (Fig. 5B, right). For example, VlsE, IR4, and IR6 displayed the highest antigenicity ($P = 6.8e-10$, $1.2e-7$, and $2.1e-8$, respectively, with an overall $P = 2.2e-10$ by ANOVA). Antigenicity of the IR1, IR2, and IR5 peptides against the rabbit polyclonal antibodies did not differ or differed weakly from that of BSA, the negative control ($P = 0.855$, 0.011, and 0.236 by ANOVA, respectively). The preimmunized rabbit sera did not react with any antigens (unpublished data; $P = 1.76e-6$ by paired $t$ test between the preimmunized and postimmunized sera).

In sum, these ELISA results suggested that (i) anti-VlsE antibodies were present in patients throughout different stages of Lyme disease, (ii) antibodies against the VlsE IRs were strongly present in naturally infected or artificially immunized nonreservoir hosts but minimally present in reservoir hosts, and (iii) the IR4 and IR6 peptides were highly immunogenic conserved epitopes on the VlsE molecule in nonreservoir hosts relative to the IR1, IR2, and IR5 peptides. These results were consistent with the conclusions of earlier studies on the antigenicity of VlsE and conserved epitopes, which established the use of VlsE and the C6 peptide (derived from IR6) in both the standard and modified diagnostics tests for Lyme disease (18, 28, 29, 42–46).

Here, we established that the IR4 peptide was as antigenic as the IR6 peptide. Indeed, both IR peptides reacted at a level similar to the reactivity of the whole VlsE protein with the sera from naturally infected and immunized hosts (Fig. 5). The use of the highly conserved IR4 and IR6 as targets for theragnostic agents has the advantage that they are expected to exhibit antigenicity against a broad set of *B. burgdorferi* strains, with the potential to mitigate the challenge of strain-specific antigenicity of the highly variable antigens including VlsE and OspC (47, 48).

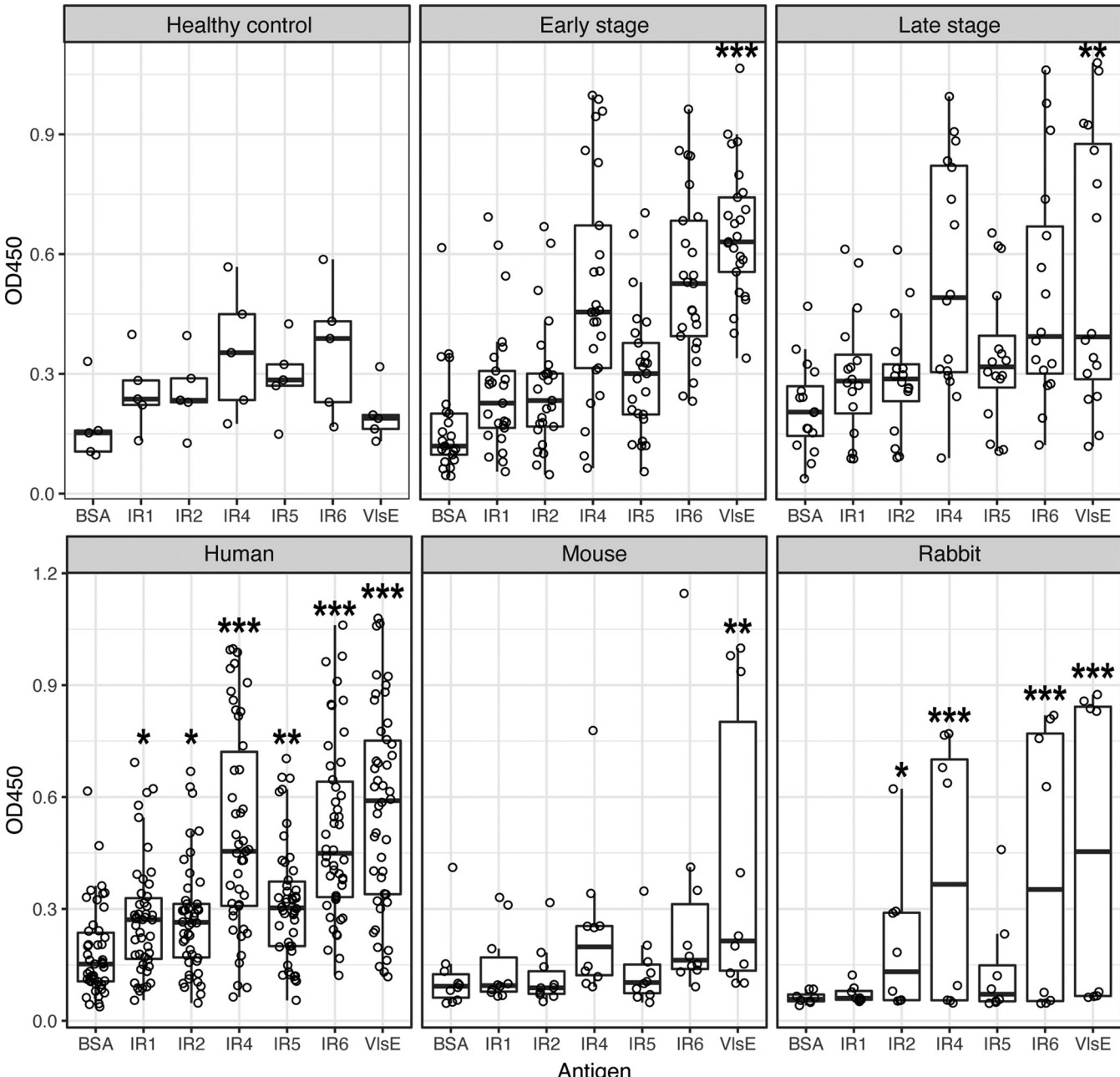

**FIG 5** Antigenicity of conserved epitopes against host sera. The antigenicity of five IRs (*x*-axis) was quantified with ELISA (see Materials and Methods). Bovine serum albumin (BSA) was used as the negative control and the purified recombinant VlsE protein (of strain B31) as the positive control. Each IR peptide was tested for reactivity (optical density at 450 nm [OD$_{450}$], *y*-axis) with host sera (represented by dots). (Top 3) Reactivity with 46 human sera, including those from four healthy controls (left), 25 early-stage Lyme disease patients (middle), and 17 late-stage Lyme disease patients (right). The antigens reacted significantly stronger with the patient sera than with the control sera for both the early-stage and late-stage samples (see Results for *t* test results). Reactivity of individual IRs, however, was not significant between the patient and control. VlsE reacted strongly with both the early and late-stage patient samples relative to the control sera. (Bottom) Reactivity with 46 patients (left), 10 naturally infected reservoir hosts (white-footed mouse, *Peromyscus leucopus*) (middle), and 4 New Zealand rabbits (*Oryctolagus cuniculus*) immunized with recombinant VlsE (right). Asterisks indicate levels of statistical significance: *, $0.01 < P < 0.05$; **, $0.001 < P < 0.01$; ***, $P < 0.001$.

**Identification and characterization of recombinant IR-specific monoclonal antibodies.** Recombinant VlsE of the strain B31 was overexpressed, purified, and used to immunize New Zealand rabbits (Fig. 6, gel image). IR-specific antibodies were identified via B cell sorting and by testing the reactivity of the supernatant of the 20 B cell lines against the five IR peptides with ELISA. We found that one cell line (1D11) bound specifically to the IR6 peptide and five cell lines (7C9, 15E2, 17A8, 28D3, and 42G10)

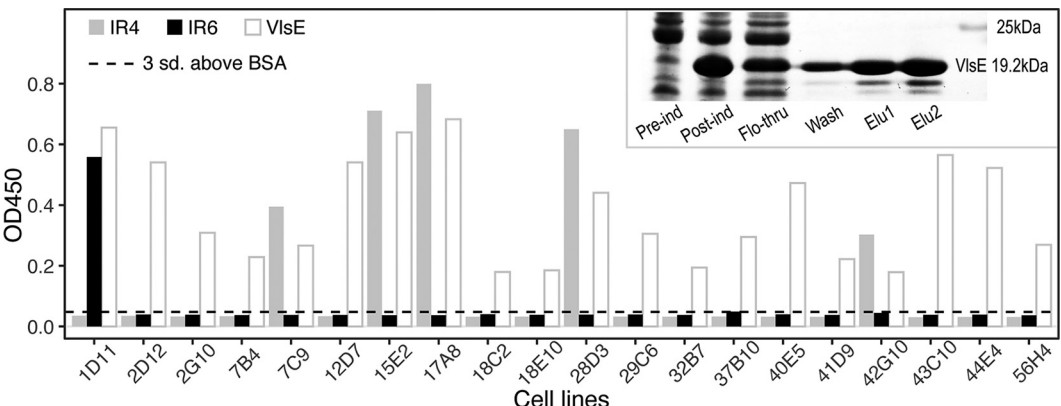

**FIG 6** Identification of rabbit B cells producing IR-specific antibodies. (Bar plot) Twenty VlsE-positive cell lines (*x*-axis) from rabbits immunized with recombinant VlsE were selected with the B cell sorting technique and tested against purified antigens with ELISA. Bovine serum albumin (BSA) was used as the negative control and the purified recombinant VlsE protein (of strain B31) as the positive control. An $OD_{450}$ value (*y*-axis) greater than 3 standard deviations above the mean BSA reactivity (dashed lines) was considered to show significant antibody-antigen reactivity. Five cell lines expressing anti-IR4 antibodies and one cell line expressing anti-IR6 antibodies were identified. (Inset) SDS-PAGE image of induction and purification of the recombinant B31 VlsE. Elution 1 and Elution 2 were combined into a single preparation with an estimated purity of ~65% and a concentration of ~5.0 mg/mL, which was subsequently used to immunize New Zealand rabbits. "Pre-ind," pre-IPTG induction; "Post-ind," postinduction; "Flo-through," flowthrough sample; "Wash," wash sample; "Elu1" and "Elu2," elution samples.

specifically to the IR4 peptide in addition to their binding to the recombinant VlsE protein (Fig. 6, bar plots). Supernatants of the remaining 14 B cell lines reacted with VlsE but not with the IR peptides, suggesting that most B cell lines in the immunized rabbit expressed antibodies recognizing epitopes located on the variable but not the conserved regions.

One pair of the most abundant heavy chain and light chain variable region ($V_H$ and $V_L$) sequences in each of four IR-specific cell lines – including the anti-IR6 1D11 cell line and three top anti-IR4 cell lines – were identified by pyrosequencing and subsequently cloned and overexpressed. Specificities of the purified recombinant monoclonal antibodies (rMAb) were validated using ELISA. The initial rMAb cloned from the 1D11 cell line based on the most abundant $V_H$ and $V_L$ sequences was not reactive to the IR6 peptide as the supernatant of the cell line did. A new rMAb – based on the second most abundant $V_H$ and $V_L$ sequences – was recloned and overexpressed and reacted with the IR6 peptide strongly and specifically. The $V_H$ and $V_L$ sequences of the four IR-specific rMAbs are shown in Table 2 and their binding characteristics were obtained by titration experiments (Fig. 7).

## DISCUSSION

**Rapid adaptive diversification of *vls* cassettes.** The *vls* gene system in *Borreliella* was discovered based on gene sequence homology with the *vsp/vlp* (variable small and large proteins) system in *Borrelia* spirochetes causing relapsing-fever (13, 14). Since then, the molecular mechanism of segmental recombination between the expression site and the archival cassettes has been well characterized in *B. burgdorferi* B31, the strain type (16, 40, 41, 49–51). In parallel, genome-based comparative analysis of the *vls* system among *Borreliella* species and strains of the same species uncovered rapid evolution in sequence, copy number, and genomic location of the *vls* cassettes (8, 33, 34).

In the present study, we showed pervasive phylogenetic inconsistencies between the *vls* gene tree and the genome-based strain tree, suggesting frequent gene duplications, gene losses, and gene exchanges, in addition to adaptive sequence evolution at the locus (Fig. 2). The highly divergent *vls* cassette sequences between phylogenetic sister strains are reminiscent of the rapid amino-acid sequence diversification at the locus encoding the outer surface protein C (*ospC*), another immunodominant antigen of *B. burgdorferi* (52). Protein sequences of major *ospC* alleles diverge in a strain-specific fashion with an average sequence identity of ~75.9% among *B. burgdorferi* strains in

**TABLE 2** Specificity and sequences of IR-specific MAbs

| MAb | Specificity | EC₅₀ (ng/mL)[a] | Variable heavy chain (V_H) sequence[b] | Variable light chain (V_L) sequence |
|---|---|---|---|---|
| 1D11-4 | IR6 | 59.07 (1.75) | MGWSCIILFLVATATGVHSQSLVESGGGLVQPEGSLTLT CKASGFSFSSGYDMCWVRQAPGKGLEYIACIDAG DDITHYASWVKGRFTVSKTSSTTVTLQLNSLT VADTATYFCGRFWDLWGPGTLVTVSS (131 aa) | MGWSCIILFLVATATGVHSSVLTQTPSPVSAAVGGTVT INCQSSQSVYDSTWLGWYQQKPGQPPKLLIYKASNL ASGVPSRFKGSGSGTHFTLTISDLECDDAATYYCVG GYSGSVDNWAFGGGTEVVVK (130 aa) |
| 15E2-1 | IR4 | 125.17 (5.10) | METGLRWLLLVAVLKGVQCQSLEESGGGLFKPT DTLTLTCTVSGIDLSSYAMIWVRQAPGKGLEWI GYIWSSGRIWYASWAKGRFTISRTSTTVDLKL ASPTTEDTATYFCARLWDIWGPGTLVTVSS (128 aa) | MDTRAPTQLLGLLLLWLPGATFAQVLTQTPSSVSAAVGG TVTISCQASQSLYNGVNLAWYQQKPGQPPKLLIFGASNL ESGVSSRFRGSGSGTQFTLTISGVQCDDAATYYCLGEF SCSSADCLAFGGGTEVVVK (135 aa) |
| 17A8-1 | IR4 | 86.14 (2.44) | METGLRWLLLVAVLKGVQCQSVEESGGRLVTPGTP LTLTCTVSGFPLSSYSMAWVRQAPGKGLEYIGFIN TDGSAYYASWAKGRITISKTSTTVELKITSPTTED TATYFCGTGNIWGPGTLVTVSS (127 aa) | MDTRAPTQLLGLLLLWLPGATFAQVLTQTPSSVSAAVGGTVTI NCQASQSVSNNNVLAWFQQKPGQPPKRLIYSALTLDSGV PSRFKGSGSGTHFTLTISGVQCDDAATYYCAGGYDCSSN DCIAFGGGTEVVVK (135 aa) |
| 28D3-1 | IR4 | 245.91 (7.93) | METGLRWLLLVAVLKGVQCQSVEESGGRLVTPGTPL TLTCTVSGFSLSSYSMGWVRQAPGKGLEYIGMII SNNSTYYASWAKGRITISKTSTTVELKITSPTTED TATYFCGTGNIWGPGTLVTVSS (127 aa) | MDTRAPTQLLGLLLLWLPGATFAQVLTQTPASVSAAV GGTVTINCQASQSTSNNNALAWFQQKPGQPP KRLIYSALTLDSGVPSRFKGSGSGTHFTLTISGVQCDDAA TYYCAGGYDCSSNDCITFGGGTEVVVK (135 aa) |

[a]EC₅₀, effective concentration of 50% response level (and standard error) based on titration by ELISA. A lower EC₅₀ indicates effective binding at a lower concentration (Fig. 7).
[b]Peptide sequences from the topmost (or 2nd topmost) abundant sequences identified in the sorted single B cells producing IR-specific antibodies. The B cells originated from rabbits immunized with recombinant VlsE proteins (see Materials and Methods).

the Northeast US, due to a history of recombination among coexisting strains and diversifying selection driven by host immunity and possibly host-species preferences (53–56). In contrast, the coding sequences of the *vls* cassettes vary at a significantly higher level between the eight major sequence clusters (~56.3% average sequence identity), while varying at a high level between copies within the same genome as well (e.g., 81.0% for B31, 76.5% for N40, and 76% for JD1) (Fig. 2). Such a high sequence variability among the *vls* cassettes is caused by elevated intrinsic mutation and recombination rates mediated by site-specific genetic mechanisms including error-prone repair and frequent gene conversion (41, 49, 57). Nevertheless, positive natural selection, by which new VlsE sequence variants are advantageous to survive the host adaptive immunity, is fundamental and essential for maintaining the high sequence variability among vls cassettes (34). locus

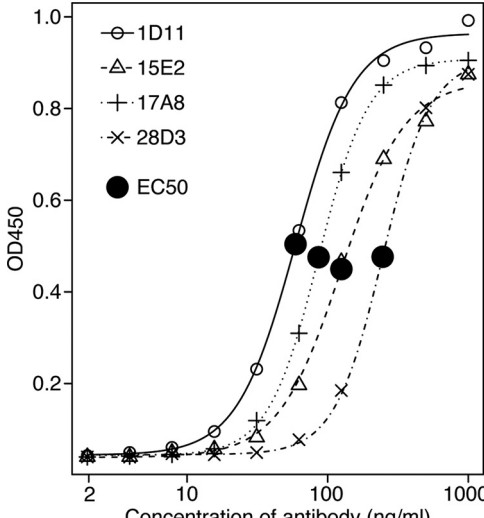

**FIG 7** Binding characteristics of IR-specific monoclonal antibodies. Serially diluted preparations of the four affinity-purified IR-specific rMAbs cloned from rabbits immunized with recombinant VlsE were tested with ELISA against their respective IRs (1D11 against IR6; 15E2, 17A8, and 28D3 against IR4). The R package *drc* (Version 3.0-1) (79) was used to estimate the effective concentration and to plot the titration curves. EC50 (effective concentration at 50% of the maximum activity) values were estimated from the fitted curves, with a lower EC50 indicating stronger antigen affinity.

As more *Borreliella* genomes are sequenced, the bioinformatics workflow including the customized web-based tool (http://borreliabase.org/vls-finder) established in the present study will facilitate large-scale automated identification of *vls* sequences and quantification of the rates of gene duplication, losses, exchanges, and sequence divergence in this key adaptive molecular system in *Borreliella*.

**Immunogenicity of the IRs in nonreservoir hosts.** The VlsE and its derivative C6 peptide (based on IR6) are key diagnostic antigens in serological tests of Lyme disease (21, 28, 58). In the present study, we confirmed the predominant immunogenicity of IR6 in serum samples from human patients and VlsE-immunized rabbits (Fig. 5). In contrast to an earlier study but consistent with another one (18, 23), the IR4 peptide showed as a similar level of antigenicity as the IR6 peptide in all three host species. Indeed, epitopes on IR4 might be more immunodominant than the IR6 epitopes in rabbits, as we obtained five anti-IR4 cell lines and only one anti-IR6 cell line out of a total of 20 randomly selected VlsE-reactive B cell lines (Fig. 6). The IR1, IR2, and IR5 appeared to be barely immunogenic in reservoir hosts as well as nonreservoir hosts (Fig. 6).

Epitope mapping studies suggested that the IR6 may function as a single conformational epitope (43). On an intact VlsE molecule (or its dimerized structure), the IR6 is almost entirely buried underneath the membrane surface and immunofluorescence assays demonstrated that the IR6 was inaccessible to antibodies on intact spirochetes (22, 24, 59). It appears paradoxical that IR4 and IR6, two highly conserved and mostly buried regions on VlsE, contain immunodominant epitopes in human patients. Evolutionary arms races drive codiversification of the antigen sequences in microbial pathogens along with the sequences of antigen-recognition proteins in vertebrate hosts through population mechanisms like negative frequency-dependent selection (55, 60, 61). Regions on antigen molecules shielded from host immune systems, like the IRs on VlsE, are not under such diversifying selection and thereby expected to be conserved in molecule sequences. The paradox resolves itself however when one considers that the IRs were indeed weakly immunogenic in the infected mice that belong to the natural reservoir species of *B. burgdorferi* (Fig. 6). Indeed, as the whole VlsE molecule elicits significant antibody responses in the infected *P. leucopus* mice, such immunogenicity is likely due to epitopes on the variable regions as expected from the pathogen-host coevolutionary arms race (44) (Fig. 6).

Likely, *B. burgdorferi* is well adapted to natural hosts and able to maintain a high level of cell integrity including intact VlsE molecules on the cell surfaces. Indeed, *B. burgdorferi* expresses cell surface proteins binding specifically to proteins of the host complement system to downregulate innate and adaptive host immunity (31, 62–64). On an intact spirochete cell surface, the VlsE molecules can shield other surface antigens from being recognized by antibodies (46).

In nonnatural hosts, such as humans and rabbits to which *B. burgdorferi* is poorly adapted, the pathogen may lose or diminish its ability to inhibit host immune responses and is thus more easily recognized by the host immune system. Upon cell disintegration and degradative processing of the surface antigens including VlsE by the major histocompatibility complex (HMC), the IR6 would be exposed along with other epitopes and elicit strong antibody responses. Because the IRs are conserved among the *vls* alleles and, unlike the VRs, their total amount remains stable during antigenic shift during infection, the IRs would result in stronger and more long-lasting host responses and become immunodominant in nonnatural hosts including humans.

**Potential of rMAbs as diagnostic agents.** *B. burgdorferi* infection is characterized by a low number of colonizing spirochetes. It is difficult to directly detect the pathogen through culture or PCR approaches due to the extreme scarcity of the organism in infected hosts (58). Further, current standard diagnostic assays of Lyme disease, targeting the anti-VlsE or anti-C6 antibodies, do not distinguish between active and past infections (25, 29). The VlsE-recognizing recombinant monoclonal antibodies are potential diagnostic agents for the direct detection of *B. burgdorferi* infections. For example, we plan to label the IR-specific rMAbs with a radioactive isotope, such as zirconium-89, and perform a positron emission tomography (PET) for the sensitive detection of trace quantities of spirochetes in experimentally infected mice. Clinical immunoPET has been shown to detect very small lesions

(~1 mm³) bearing cancer antigens with very low density (<500 antigen copies per cell) (65). As a result, we are hopeful that these rMAbs can be harnessed for the immunoPET of Lyme disease despite the scarcity of spirochetes during human infection, but only rigorous preclinical experiments will show us whether this is possible.

To validate the utility of these rMAbs as diagnostic agents, it is necessary to perform *in vitro* testing using cultured *B. burgdorferi* cells followed by *in vivo* testing using a mouse model of Lyme disease. We anticipate several biological and technical challenges during *in vitro* and *in vivo* validation testing of the rMAbs. For *in vitro* testing using cultured spirochetes, first, it is unclear if the rMAbs would bind VlsE anchored on the surface of live *B. burgdorferi* cells because of limited surface accessibility of the IRs at native conformations, even though the rMAbs reacted strongly with VlsE molecules fixed on an ELISA plate (Fig. 2) (43, 44). Molecular conformation may also differ between the IR peptides used in ELISA and the IRs as a part of VlsE molecules anchored to the outer membrane of spirochete cells. Second, *B. burgdorferi* does not constitutively express a large quantity of VlsE during *in vitro* culture, and supplementing the standard media with human tissue cells may be necessary to increase VlsE expression for *in vitro* validation of rMAb binding (66, 67). Third, both the IR4 and IR6 sequences vary slightly among *B. burgdorferi* strains despite high sequence conservation (Fig. 3). The affinity of these rMAbs, which were raised using a single allelic variant (the B31 VlsE), is expected to vary among the *B. burgdorferi* strains. Effects of sequence variability to rMAb affinity could be quantified with ELISA using synthetic peptides representing the IR variants. Ideally, amino acid residues essential for the rMAb binding could be accurately pinpointed with systematic epitope mapping (23).

## MATERIALS AND METHODS

**Identification of *vls* cassette sequences and evolutionary analysis.** We downloaded the whole-genome sequences of 13 *B. burgdorferi* strains from NCBI GenBank (35). The silent cassette sequences of the B31-5A3 clone (GenBank accession U76406) (14) were used as the queries to search for sequences homologous to the *vls* cassette sequences using HMMER (version 3.3.2) (68). A customized web-based software tool was developed to identify and extract individual *vls* sequences given a *B. burgdorferi* replicon sequence (http://borreliabase.org/vls-finder). Sequences of the silent cassettes and *vlsE* were translated, aligned, and converted into a codon alignment using MUSCLE (version 3.8.31) (69) and the *bioaln* utility (–dna2pep method) of the BpWrapper (version 1.13) toolkit (70). A maximum likelihood tree was subsequently inferred using IQ-TREE (version 1.6.1) with the best-fit nucleotide substitute model KOSI07 and 1000 bootstrap replicates (71). Branches with lower than 80% bootstrap support were collapsed using the *biotree* utility (-D method) of the BpWrapper toolkit (70). The tree was rendered using the R package *ggtree* (Version 2.2.4) (72). To quantify the sequence conservation, evolutionary rates at individual amino acid positions were estimated using Rate4Site (version 3.0.0) with the protein alignment and the phylogenetic tree as inputs and the B31-5A3 VlsE sequence (GenBank accession U76405) as the reference (73). Sequence conservation at the IRs was further quantified and visualized with WebLogo (74).

**Synthesis of peptides representing conserved epitopes of VlsE.** The preparation of the peptides was based on the annotation of the B31-5A3 VlsE protein sequence (14). Five invariant regions, IR1, IR2, IR4, IR5, and IR6, were tested for antigenicity using sera from three host species. IR3 (AGKLFVK), the shortest IR, was excluded from the antigenicity test. Extra flanking amino acids were added to IR2, IR4, and IR5 to meet the minimum length for peptide synthesis. Peptides were commercially synthesized and biotin-labeled on the N terminus using Fmoc chemistry (GenScript, Piscataway, NJ, USA). Sequences of these peptides are shown in Table 1.

**Sera collection from naturally infected hosts.** The 56 serum samples, consisting of Lyme patient and control sera provided by the US Centers for Disease Control and Prevention (CDC; n = 40), Lyme patient sera provided by Maria Gomes-Solecki (University of Tennessee Health Science Center; n = 6), and sera from naturally infected individuals of the reservoir host white-footed mouse (*Peromyscus leucopus*) originated from Millbrook, New York (n = 10), have been used and described in previous publications (53, 75, 76). Briefly, among the human samples, 25 serum samples were derived from patients with early-stage Lyme disease including those diagnosed as having the skin symptom erythema migran (EM) or as EM convalescence. Seventeen human serum samples were from patients displaying late-stage Lyme disease symptoms including arthritic, cardiac, and neurological Lyme diseases. Four human serum samples were from healthy individuals as controls.

**Cloning, overexpression, and purification of recombinant VlsE protein.** Recombinant VlsE protein from the B31 strain was cloned, overexpressed, and purified using a protocol described previously (53). Briefly, the 585-bp *vlsE* cassette region (including the IR1 through VR6 regions) of the B31-5A3 clone was codon-optimized, synthesized, and cloned into the pET24 plasmid vector which then transfected *Escherichia coli* BL21 cells. A10 × Histidine-tag was added on the N terminus of the construct to facilitate the downstream purification. All cloning work was performed by a commercial service (GeneImmune

Biotechnology Corp., Rockville, MD, USA). The *E. coli* strain that contained a cloned *vlsE* cassette was cultured in Luria-Bertani (LB) broth containing 0.4% glucose and 50 $\mu$g/mL Ampicillin. When the culture reached exponential growth, we induced the expression of the cloned *vlsE* cassette by adding isopropyl $\beta$-d-1-thiogalactopyranoside (IPTG) to a final concentration of 0.25 mM and by incubation overnight at 25°C. Cells were collected and then lysed by lysozyme and sonication. The lysate supernatant, containing the recombinant VlsE protein, was purified using nickel Sepharose beads (Ni-NTA, Thermo Fisher Scientific, Waltham, MA, USA) following the manufacturer's protocol. The identity and concentration of the purified protein were examined and quantified using the sodium dodecyl sulfate-polyacrylamide gel electrophoresis (SDS-PAGE) and the Pierce Bradford Protein assay kit (Thermo Fisher Scientific, Waltham, MA, USA).

**Immunization of rabbits and preparation of polyclonal and monoclonal antibodies.** Antibody preparation was conducted with a commercial service GenScript (Piscataway, NJ, USA). Briefly, the project consisted of four stages. In stage 1, animals were immunized, and polyclonal antibodies were obtained. Specifically, four New Zealand rabbits (*Oryctolagus cuniculus*) were immunized with 100 $\mu$g purified recombinant VlsE protein on days 1, 14, and 28. The rabbits were bled for antiserum collection 1 week after the third immunization. The antisera were subsequently purified by affinity chromatography to obtain polyclonal antibodies (pAbs), which were assayed for anti-VlsE activity. In stage 2, monoclonal antibodies (MAbs) were identified via single B cell sorting. Peripheral blood mononuclear cells (PBMC) were collected from the two selected immunized rabbits 1 week after a booster dose with the recombinant VlsE. Plasma B cells (CD138$^+$) were isolated and enriched using a commercial kit. B cells were transformed by a proprietary process and then cloned by limiting dilution. The supernatants of positive cell lines were used to test for binding with VlsE and positive cell lines were chosen to produce monoclonal antibodies. In Stage 3, the variable domains of the light and heavy chains of the VlsE-binding antibodies were sequenced. Total RNA was isolated from the VlsE-binding B cell lines and reverse-transcribed into cDNA using universal primers. DNA sequences encoding variable domains of the heavy chain and the light chain were amplified and sequenced. In Stage 4, the amplified antibody variable fragments were cloned into plasmid vector pcDNA3.4, which was then transfected into mouse cells for expression. Supernatants of cell cultures were harvested continuously. The recombinant monoclonal antibodies (rMAbs) were purified using protein A/G affinity chromatography (with immobilized protein A and G from *Staphylococcus aureus*) followed by size exclusion chromatography (SEC).

**Identification of IR-specific MAbs with ELISA.** Sera from naturally infected humans and *P. leucopus* were tested to evaluate reactivity to the IR peptides (Table 1) and the recombinant VlsE protein with ELISA using a protocol described previously (53). Briefly, a 96-well MICROLON 600 plate (USA Scientific, Inc., Ocala, FL, USA) was incubated with 10 $\mu$g/mL of antigen overnight at 4°C. Serum samples diluted between 1:100 to 1:1000 were applied after blocking with 5% milk and were incubated for 2 h at 37°C, followed by the application of horseradish peroxidase (HRP)-conjugated secondary antibodies. We used the Goat Anti-Human IgG/IgM (H+L) (Abcam, Cambridge, UK) 1:40,000 for assays of human sera and the Goat anti-*P. leucopus* IgG (H+L) (SeraCare Life Sciences, MA, USA) 1:1000 for assays of *P. leucopus* sera. The antigen-antibody reaction was probed by TMB ELISA Substrate Solution (Invitrogen eBioscience) and was terminated with 1 M sulfuric acid after 15 min. Binding intensities were measured at the 450 nm wavelength using a SpectraMax i3 microplate reader (Molecular Devices, LLC, CA, USA).

The same ELISA protocol was followed to test against binding with the rabbit-originated antibodies as well, including the purified anti-VlsE pAbs, the supernatants of selected B cell cultures, and the purified rMAbs. Mouse Anti-Rabbit IgG Fr secondary antibody (GenScript, Piscataway, NJ) 1:30,000 was used for assays of these rabbit-derived samples. Serial dilutions of MAbs by factors from 1,000 to 512,000 were tested with ELISA to quantify the binding activities.

**Protein structure visualization.** The PDB file of the VlsE protein structure (accession no. 1L8W) was downloaded from the protein data bank (PDB) (19). The PDF file describes a tetramer of VlsE. We used Chimera (version 1.15) (77) to visualize the protein structure in ribbon and surface-filled formats and to color the six invariable regions (IR1-6).

**Animal care.** Antibody production from the New Zealand rabbits followed the protocols approved by the Office of Laboratory Animal Welfare (OLAW) Assurance and the Institutional Animal Care and Use Committee (IACUC) of the vendor (GenScript, Piscataway, NJ).

**Data availability.** Data visualization and statistical analysis were performed in the R statistical computing environment (78) accessed with RStudio. The alignment of translated *vls* sequences, ELISA readings, and R scripts are publicly available on GitHub at https://github.com/weigangq/vls-mabs.

## ACKNOWLEDGMENTS

This work was supported by the Public Health Service awards AI139782 (WGQ) from the National Institute of Allergy and Infectious Diseases (NIAID) and EB030275 (B.M.Z. and W.Q.) from the National Institute of Bio-medical Imaging and Bioengineering (NIBIB) of the US National Institutes of Health (NIH). The content of the manuscript is solely the responsibility of the authors and does not necessarily represent the official views of NIH. Li Li and Saymon Akther were supported in part by the Doctoral Program in Biology of the Graduate Center, the City University of New York.

We thank Christopher Sexton and Jeannine Petersen of the Centers for Disease Control and Prevention (CDC) and Maria Gomes-Solecki (University of Tennessee Health

Science Center) for providing serum samples from human patients and reservoir mice, respectively.

Li Li performed evolutionary analysis, protein purification, and ELISA. Li Li composed the initial draft. Lia Di developed the bioinformatics pipeline and web tool and participated in protein purification and ELISA. Saymon Akther participated in protein purification and ELISA and edited the manuscript. Brian M. Zeglis and Weigang Qiu conceived, obtained funding for, and supervised the project. Weigang Qiu and Brian M. Zeglis revised the manuscript.

We declare no conflicts of interest.

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
