## [Reviewer comments · Microbiology Spectrum]

Microbiology Spectrum

Evolution of the *vls* antigenic variability locus of the Lyme disease pathogen and development of recombinant monoclonal antibodies targeting conserved VlsE epitopes

Li Li, Lia Di, Saymon Akther, Brian Zeglis, and Weigang Qiu

Corresponding Author(s): Weigang Qiu, Hunter College of City University of New York

Review Timeline:

Submission Date:	May 10, 2022
Editorial Decision:	June 9, 2022
Revision Received:	September 2, 2022
Accepted:	September 2, 2022

Editor: Catherine Brissette

Reviewer(s): The reviewers have opted to remain anonymous.

Transaction Report:

DOI: <https://doi.org/10.1128/spectrum.01743-22>

June 9, 2022

Dr. Weigang Qiu
Hunter College of City University of New York
Department of Biological Sciences
Hunter College
695 Park Av
New York 10065

Re: Spectrum01743-22 (Intra-specific diversification of a multi-copy surface-antigen system in Lyme disease pathogens and development of recombinant monoclonal antibodies targeting conserved VlsE epitopes)

Dear Dr. Weigang Qiu:

Both reviewers feel this work has merit, but requires modifications before publication.

Link Not Available

Sincerely,

Catherine Brissette

Journals Department
Reviewer comments:

Reviewer #1 (Comments for the Author):

This manuscript describes the phylogeny of vlsE sequences and the production and properties of rabbit monoclonal antibodies against the invariant regions of VlsE of *Borrelia burgdorferi*. Several aspects that require refinement are described below.

1. The manuscript should indicate that there are several prior publications regarding experimental infection of rabbits with *B. burgdorferi*; see e.g. Crother et al. *Infect Immun.* 2004 Sep;72(9):5063-72 and Embers et al. *FEMS Immunol Med Microbiol.* 2007 Aug;50(3):421-9.

2. l. 286-287. This sentence is misleading, in that IR6 is not completely conserved among all species and strains.
3. l. 297-319. It should be stated more clearly in this section and the accompanying tables and figure legends that the sera are from Lyme disease patients and rabbits immunized with recombinant VlsE. Also, it is not indicated anywhere in the manuscript whether the *Peromyscus* mice were infected or uninfected. It is presumed that the mice were not infected and vivarium raised (not wild); if so, this should be clearly described in the Materials and Methods, Results, and Discussion section. Also, the results should be interpreted accordingly. It would be expected that uninfected *Peromyscus* would not be reactive.
4. A flaw in the current study is that reactivity of uninfected human control sera and unimmunized rabbit sera are not evaluated. Sera from uninfected *Peromyscus* deer mice should also be included, if the sera tested are from infected animals. These results are necessary to evaluate the reactivity of sera and ascertain that the sera of infected human and immunized rabbit subjects have a significantly higher reactivity (as compared to normal sera as opposed to a BSA control).
5. l. 359-367. This paragraph could also include relevant publications on vlsE recombination in B31 (Zhang and Norris *Infect. Immun.* 1998) and *B. garinii* and *B. afzelii* strains (Wang D et al. *Molec. Microbiol.* 2003).
6. It needs to be indicated in the Discussion that the current study does not provide information about potential immunoprotective activity of the rMAbs. As such, it seems to be premature to describe these antibodies as "promising theragnostic agents". Likewise, it is unnecessary and premature to describe future directions (such as the PET studies) in the final paragraph. Many methods have already been developed for detecting antibody binding to *Borrelia* organisms and for detecting *B. burgdorferi* during infection.

Minor

- l. 117-118. The phrase "As a multi-copy gene family and driven by adaptive amino-acid substitutions" is not accurate, in that the silent cassettes are not genes; also, it is unclear what is meant by "adaptive amino-acid substitutions".
- l. 199-200. Liang et al. *J. Clin. Microbiol.* 1999 should also be cited here, given that it was the first description of diagnostic utility of VR6 and VR4. Also, Bacon et al. used laboratory-produced VlsE tests, not commercial VlsE-based tests.
- l. 196. Suggested wording: B cells producing anti-VlsE monoclonal antibodies ... l. 200. It should be stated here that the B cells were transformed by a proprietary process and then cloned by limiting dilution prior to screening for anti-VlsE antibody production. l. 207. pcDNA3.4 and then transfected into...
- l. 257-261. This sentence is overly long and unclear; it should be rewritten into two sentences.
- l. 294-296. The statement is confusing because prior sentences stated that IR6 has limited surface exposure. Also, "the" prior to IR4 and IR6 should be removed.
- l. 354-356. The phrase "The VH and VL sequences of the four IR-specific rMAbs..." does not fit with the description "were obtained by titration experiments".
- l. 452-453. It is unclear what is meant by "IRs in human serum".

Reviewer #2 (Comments for the Author):

This manuscript reports on two interesting aspects of the hypervariable surface antigen, VlsE that is involved in the antigenic variation process in Lyme borreliae. The first aspect, which I believe is unique, is the report of the lack of congruence between the sequence conservation of the vls cassettes and the genome phylogenies of the *Borrelia* strains from which the vls sequences were taken. This provides interesting new information on the ongoing evolution of the vls locus, which occurs at a very high rate and by mechanism(s) that have not been elucidated. The second aspect is that the authors tested the antigenicity of IR (Invariant Region) peptides in the white-footed mouse reservoir, in human serum and in New Zealand rabbits. The IR4 and IR6 (target of the C6 peptide diagnostic assay) peptides showed strong interaction with human and rabbit sera, but not with serum from the natural mouse reservoir. Four monoclonal antibodies to IR4 and IR6 peptides were cloned and purified and are proposed as possibilities for the development of diagnostic and theragnostic agents for direct detection of *B. burgdorferi*. Other than the antibodies themselves, I am not sure that there is very much here that has not been previously reported.

The work reported appears to have been well-designed and executed and provides some interesting new information. However, I have a number of concerns regarding the manuscript in terms of explanation of the results and their implications:

- 1) The first line of the title is not clear or informative. The title needs to be rewritten to be understandable to most readers.
- 2) The authors state that the inconsistencies of the sequence phylogeny with the genome phylogeny indicate rapid gene duplications, losses and recombination at the *vls* locus. However, how they reached this conclusion is not clear. Their data indicate clear differences in sequence, but I don't see the type of granular analysis needed to arrive at their conclusions.
- 3) The authors show a very high rate of sequence variation at the *vls* locus. Can their data shed light on how this might occur? For example, sequence variation can be driven by increased mutation frequency (eg, mutator strains) or by stringent diversifying selection or both. Is the immune selection likely to be enough? If the authors do indeed have data supporting rapid accumulation of gene duplications, losses and recombination, then perhaps some mechanism is at play at the DNA level to promote increased levels of mutations of these types in the *vls* locus. Some discussion of this possibility would be helpful.
- 4) For the 2nd major aspect of the manuscript, the antibodies and antigenic regions, I am a bit unclear about what aspects of this report are unique to this work and not previously reported in the large number of papers previously published on the subject in the past few decades. This should be clarified by the authors to strengthen the significance of their own work versus whether it is primarily a simple repeat of previously published findings.
- 5) The authors state in the Abstract (lines 40-42) and Importance Statement (lines 48-50) that that the IR-specific monoclonals are promising diagnostic and theragnostic agents. But this statement requires a very large leap of faith based upon the preliminary data reported here and the lack of clear explanation as to how the monoclonals would be used to be more sensitive than PCR.
- 6) Line 48 calls the monoclonals a novel class. What is novel about them?
- 7) The introduction of the antigenic variation system should incorporate the reviews by Norris (2014) and Chaconas et al (2020) on line 75.
- 8) Line 135 and elsewhere - the term *vlsS* is not commonly used, probably because it is confusing. It is a singular term but denotes 15 silent cassettes (in the B31 strain). I would suggest replacing *vlsS* with "silent cassettes" or "cassettes 2-16".
- 9) Line 292 - the two papers that provide the only evidence that *VlsE* functions as a dimer need to be added here (Verhey et al 2018 Mol Micro and Verhey et al 2019).
- 10) Line 363 - two important references on the mechanism of the gene conversion events in B31 should be added here (Verhey et al 2018, Mol Micro and Verhey et al 2018, Cell Reports).
- 11) Line 383 - One could argue that it is not simply a more intense immune selection, but that the types of mutagenic events observed in *vlsE* suggest the possibility of a different and more high frequency mechanism for generating DNA changes.
- 12) More than two pages of the Discussion deal with Future Work. This is superfluous and can be summed up in a closing sentence or two.
- 13) Line 439-441 does not tell us how monoclonals provide a solution to the scarcity of spirochete counts in humans.
- 14) Line 456, FYI, the reference by Bykowski et al 2006, J Bact on *VlsE* expression, provides info on maximizing *VlsE* expression in vitro [but the last section of the Discussion should nonetheless be removed anyway].
- 15) The last paragraph (line 464-469) is both speculative and confusing [but the last section of the Discussion should nonetheless be removed anyway].
- 16) Nowhere in the manuscript do I see any information on how monoclonal antibodies are going to overcome the scarcity of spirochetes in human infections and exceed the performance of PCR for detection of *B. burgdorferi*. So claims about diagnostic and theragnostic potential are unjustified.
- 17) Figure 1A is confusing. The *vlsE* locus has always been described as being at the right end of *lp28-1*. When drawn in this orientation the *vlsE* gene is expressed from left to right or in the same orientation as the protein diagram shown in Fig 1B. However, the current Fig. 1A has the gene expressed from right to left, in the opposite orientation of the protein diagram in Fig. 1B. Please flip the diagram in 1A to match orientation with 1B.

Staff Comments:

Preparing Revision Guidelines

Please return the manuscript within 60 days; if you cannot complete the modification within this time period, please contact me. If you do not wish to modify the manuscript and prefer to submit it to another journal, please notify me of your decision immediately so that the manuscript may be formally withdrawn from consideration by Microbiology Spectrum.

Reply to reviewers

Re: **Spectrum01743-22** (Intra-specific diversification of a multi-copy surface-antigen system in Lyme disease pathogens and development of recombinant monoclonal antibodies targeting conserved VlsE epitopes)

Notes to Editor

Dear Dr. Brissette, Editor, Microbial Spectrum,

Thanks for organizing the peer review for our manuscript, which was highly constructive. In the thoroughly revised manuscript, we have accepted and followed virtually all recommendations from the two referees. Please see below for our point-by-point responses.

We are sorry for a delay in returning the revision, which was due to additional experiments requested by a reviewer, analysis of the new data, extensive revision, and as well as obtaining approval from CDC for the publication of the human sera used in the present study.

Sincerely,

Weigang Qiu, Ph.D., Professor
Department of Biological Sciences
Hunter College, City University of New York
Email: wqiu@hunter.cuny.edu

Responses to Reviewer #1

This manuscript describes the phylogeny of vlsE sequences and the production and properties of rabbit monoclonal antibodies against the invariant regions of VlsE of *Borrelia burgdorferi*. Several aspects that require refinement are described below.

1. The manuscript should indicate that there are several prior publications regarding experimental infection of rabbits with *B. burgdorferi*; see e.g. Crother et al. *Infect Immun.* 2004 Sep;72(9):5063-72 and Embers et al. *FEMS Immunol Med Microbiol.* 2007 Aug;50(3):421-9.

Response: Thanks for pointing out these prior works. Both references have been added to relevant parts of the texts.

2. 1. 286-287. This sentence is misleading, in that IR6 is not completely conserved among all species and strains.

Response: The vague statement is removed and instead we refer to the precise quantitative values of conservation in Table 1.

3. 1. 297-319. It should be stated more clearly in this section and the accompanying tables and figure legends that the sera are from Lyme disease patients and rabbits immunized with recombinant VlsE. Also, it is not indicated anywhere in the manuscript whether the *Peromyscus* mice were infected or uninfected. It is presumed that the mice were not infected and vivarium raised (not wild); if so, this should be clearly described in the Materials and Methods, Results, and Discussion section. Also, the results should be interpreted accordingly. It would be expected that uninfected *Peromyscus* would not be reactive.

Response: We qualified rabbit sera as obtained by recombinant VlsE in all texts, table captions, and figure legends. We clarified with our collaborator (Dr Maria Gomes-Solecki) who provided us the deer mouse sera that the 10 deer mice were collected from the field and naturally infected as validated by C6-based immunoassays (not vivarium-raised or artificially immunized). The infection status of the deer mice sera was added to the M&M and the Fig 5 caption.

4. A flaw in the current study is that reactivity of uninfected human control sera and unimmunized rabbit sera are not evaluated. Sera from uninfected *Peromyscus* deer mice should also be included, if the sera tested are from infected animals. These results are necessary to evaluate the reactivity of sera and ascertain that the sera of infected human and immunized rabbit subjects have a significantly higher reactivity (as compared to normal sera as opposed to a BSA control).

Response: Uninfected human control sera were included but mixed with infected sera in the previous Fig 5. We now plot the control sera separately to highlight the difference. We performed a new set of ELISA, which validated that the unimmunized rabbit sera were indeed not reactive. The related statements were modified accordingly.

5. 1. 359-367. This paragraph could also include relevant publications on vlsE recombination in B31 (Zhang and Norris Infect. Immun. 1998) and *B. garinii* and *B. afzelii* strains (Wang D et al. Molec. Microbiol. 2003).

Response: The two references were added.

6. It needs to be indicated in the Discussion that the current study does not provide information about potential immunoprotective activity of the rMAbs. As such, it seems to be premature to describe these antibodies as "promising theragnostic agents". Likewise, it is unnecessary and premature to describe future directions (such as the PET studies) in the final paragraph. Many methods have already been developed for detecting antibody binding to *Borrelia* organisms and for detecting *B. burgdorferi* during infection.

Response: This seems to be a shared concern from both reviewers (see below). As such, we removed the theragnostic potential from the concluding statements (and abstract). Further, the last paragraph regarding PET studies has been deleted.

Minor

1. 117-118. The phrase "As a multi-copy gene family and driven by adaptive amino-acid substitutions" is not accurate, in that the silent cassettes are not genes; also, it is unclear what is meant by "adaptive amino-acid substitutions".

Response: We added the "silent" to clarify that vls cassettes are not expressed genes. We replaced the "adaptive amino-acid substitutions" with "diversifying natural selection".

l. 199-200. Liang et al. J. Clin. Microbiol. 1999 should also be cited here, given that it was the first description of diagnostic utility of VR6 and VR4. Also, Bacon et al. used laboratory-produced VlsE tests, not commercial VlsE-based tests.

Response: We assume the sentence was on lines 99-100. We added Liang et al (1999) and removed the Bacon et al, as suggested.

l. 196. Suggested wording: B cells producing anti-VlsE monoclonal antibodies ... l. 200. It should be stated here that the B cells were transformed by a proprietary process and then cloned by limiting dilution prior to screening for anti-VlsE antibody production. l. 207. pcDNA3.4 and then transfected into...

Response: modified as suggested.

l. 257-261. This sentence is overly long and unclear; it should be rewritten into two sentences.

Response: This paragraph has been re-written, adding more details on the reasoning and a new reference on lineage sorting.

l. 294-296. The statement is confusing because prior sentences stated that IR6 has limited surface exposure. Also, "the" prior to IR4 and IR6 should be removed.

Response: IR6 was removed from this statement due to conflicting information, as suggested.

l. 354-356. The phrase "The VH and VL sequences of the four IR-specific rMAbs..." does not fit with the description "were obtained by titration experiments".

Response: The sentence was revised.

l. 452-453. It is unclear what is meant by "IRs in human serum".

Response: The sentence was revised.

Response to Reviewer #2

This manuscript reports on two interesting aspects of the hypervariable surface antigen, VlsE that is involved in the antigenic variation process in Lyme borreliosis. The first aspect, which I believe is unique, is the report of the lack of congruence between the sequence conservation of the vls cassettes and the genome phylogenies of the Borrelia strains from which the vls sequences were taken. This provides interesting new information on the ongoing evolution of the vls locus, which occurs at a very high rate and by mechanism(s) that have not been elucidated.

Response: We are glad to see the affirmation that our evolutionary analysis is novel and important.

The second aspect is that the authors tested the antigenicity of IR (Invariant Region) peptides in the white-footed mouse reservoir, in human serum and in New Zealand rabbits. The IR4 and IR6 (target of the C6 peptide diagnostic assay) peptides showed strong interaction with human and rabbit sera, but not with serum from the natural mouse reservoir. Four monoclonal antibodies to IR4 and IR6 peptides were cloned and purified and are proposed as possibilities for the development of diagnostic and therapeutic agents for direct detection of B. burgdorferi. Other than the antibodies themselves, I am not sure that there is very much here that has not been previously reported.

Response: Our primary goal for this part of report is indeed to present the four MAbs as a new resource for their potential uses in diagnostics and research. The manuscript is revised to clarify this part of our research goal. For example, in the abstract, we replace “theragnostic” with “research” and replaced “promising” with “potential novel”.

The work reported appears to have been well-designed and executed and provides some interesting new information.

Response: It’s reassuring to read this confirmation of our study design and protocol.

However, I have a number of concerns regarding the manuscript in terms of explanation of the results and their implications:

1) The first line of the title is not clear or informative. The title needs to be rewritten to be understandable to most readers.

Response: The title has been revised to be “Evolution of an antigenic-variability locus of the Lyme disease pathogen and development of recombinant monoclonal antibodies targeting conserved VlsE epitopes”

2) The authors state that the inconsistencies of the sequence phylogeny with the genome phylogeny indicate rapid gene duplications, losses and recombination at the vls locus. However, how they reached this conclusion is not clear. Their data indicate clear differences in sequence, but I don't see the type of granular analysis needed to arrive at their conclusions.

Response: We added some details to our reasoning (in Results), by laying out three possible evolutionary mechanisms causing the gene tree and genome tree inconsistency. A reference is added to increase clarity.

3) The authors show a very high rate of sequence variation at the vls locus. Can their data shed light on how this might occur? For example, sequence variation can be driven by increased mutation frequency (eg, mutator strains; No) or by stringent diversifying selection (Yes) or both. Is the immune selection likely to be enough? If the authors do indeed have data supporting rapid accumulation of gene duplications, losses and recombination, then perhaps some mechanism is at play at the DNA level to promote increased levels of mutations of these types in the vls locus. Some discussion of this possibility would be helpful.

Response: We added the mechanisms of elevated mutation and recombination rates at the vls locus (including references). We concluded that both hyper-mutability and natural selection have contributed to the vls hyper-sequence variability.

4) For the 2nd major aspect of the manuscript, the antibodies and antigenic regions, I am a bit unclear about what aspects of this report are unique to this work and not previously reported in the large number of papers previously published on the subject in the past few decades. This should be clarified by the authors to strengthen the significance of their own work versus whether it is primarily a simple repeat of previously published findings.

Response: New references were added (thanks to the reviewers’ recommendations) to clarify prior work, including e.g., the discovery of antigen shifts using immunized rabbits and the discovery of IR4 and IR6 antigenicity in humans and mice.

5) The authors state in the Abstract (lines 40-42) and Importance Statement (lines 48-50) that that the IR-specific monoclonals are promising diagnostic and theragnostic agents. But this

statement requires a very large leap of faith based upon the preliminary data reported here and the lack of clear explanation as to how the monoclonals would be used to be more sensitive than PCR.

Response: We removed the parts regarding MAb potential as theragnostic agents. See response to #16 below for comparison with PCR.

6) Line 48 calls the monoclonals a novel class. What is novel about them?

Response: We removed the “class of” from this statement, which now simply says “novel MAbs that have the potential to....”

7) The introduction of the antigenic variation system should incorporate the reviews by Norris (2014) and Chaconas et al (2020) on line 75.

Response: These two references have been added here.

8) Line 135 and elsewhere - the term vlsS is not commonly used, probably because it is confusing. It is a singular term but denotes 15 silent cassettes (in the B31 strain). I would suggest replacing vlsS with "silent cassettes" or "cassettes 2-16".

Response: “vlsS” is no longer mentioned and has been replaced as suggested.

9) Line 292 - the two papers that provide the only evidence that VlsE functions as a dimer need to be added here (Verhey et al 2018 Mol Micro and Verhey et al 2019).

Response: These two references have been added (along with another one mentioned by the first reviewer)

10) Line 363 - two important references on the mechanism of the gene conversion events in B31 should be added here (Verhey et al 2018, Mol Micro and Verhey et al 2018, Cell Reports).

Response: These two references have been added

11) Line 383 - One could argue that it is not simply a more intense immune selection, but that the types of mutagenic events observed in vlsE suggest the possibility of a different and more high frequency mechanism for generating DNA changes.

Response: The statement was modified to include the accelerated mutation and recombination rates.

12) More than two pages of the Discussion deal with Future Work. This is superfluous and can be summed up in a closing sentence or two.

Response: This part has been thoroughly revised and greatly simplified.

13) Line 439-441 does not tell us how monoclonals provide a solution to the scarcity of spirochete counts in humans.

Response: Please see response to #16 below.

14) Line 456, FYI, the reference by Bykowski et al 2006, J Bact on VlsE expression, provides info on maximizing VlsE expression in vitro [but the last section of the Discussion should nonetheless be removed anyway].

Response: We added this reference because it adds to the demonstration of prior work on vlsE regulation. It is also important for our future experimental design.

15) The last paragraph (line 464-469) is both speculative and confusing [but the last section of the Discussion should nonetheless be removed anyway].

Response: This part (on future work) has been greatly simplified.

16) Nowhere in the manuscript do I see any information on how monoclonal antibodies are going to overcome the scarcity of spirochetes in human infections and exceed the performance of PCR for detection of *B. burgdorferi*. So claims about diagnostic and theragnostic potential are unjustified.

We removed the theragnostic potential and added the following clarification (with a new reference) “In nuclear oncology, clinical immunoPET has been shown to detect very small lesions ($\sim 1 \text{ mm}^3$) bearing cancer antigens with very low density (< 500 antigen copies per cell). As a result, we are hopeful that these antibodies can be harnessed for the immunoPET of Lyme disease despite the scarcity of spirochetes during human infection, but only rigorous preclinical experiments will show us whether this is possible.”

17) Figure 1A is confusing. The *vlsE* locus has always been described as being at the right end of lp28-1. When drawn in this orientation the *vlsE* gene is expressed from left to right or in the same orientation as the protein diagram shown in Fig 1B. However, the current Fig. 1A has the gene expressed from right to left, in the opposite orientation of the protein diagram in Fig. 1B. Please flip the diagram in 1A to match orientation with 1B.

Response: the diagram has been flipped as recommended.

September 2, 2022

Dr. Weigang Qiu
Hunter College of City University of New York
Department of Biological Sciences
Hunter College
695 Park Av
New York 10065

Re: Spectrum01743-22R1 (Evolution of the *vls* antigenic variability locus of the Lyme disease pathogen and development of recombinant monoclonal antibodies targeting conserved VlsE epitopes)

Dear Dr. Weigang Qiu:

Your manuscript has been accepted, and I am forwarding it to the ASM Journals Department for publication. You will be notified when your proofs are ready to be viewed.

Sincerely,

Catherine Brissette
Editor, Microbiology Spectrum
